# Mutants of human ACE2 differentially promote SARS-CoV and SARS-CoV-2 spike mediated infection

**Nidhi Shukla, Sarah M. Roelle, Vinicius G. Suzart, Anna M. Bruchez**[ID]**, Kenneth A. Matreyek**[ID]*

Department of Pathology, Case Western Reserve University School of Medicine, Cleveland, Ohio, United States of America

* kenneth.matreyek@case.com

## Abstract

SARS-CoV and SARS-CoV-2 encode spike proteins that bind human ACE2 on the cell surface to enter target cells during infection. A small fraction of humans encode variants of ACE2, thus altering the biochemical properties at the protein interaction interface. These and other ACE2 coding mutants can reveal how the spike proteins of each virus may differentially engage the ACE2 protein surface during infection. We created an engineered HEK 293T cell line for facile stable transgenic modification, and expressed the major human ACE2 allele or 28 of its missense mutants, 24 of which are possible through single nucleotide changes from the human reference sequence. Infection with SARS-CoV or SARS-CoV-2 spike pseudotyped lentiviruses revealed that high ACE2 cell-surface expression could mask the effects of impaired binding during infection. Drastically reducing ACE2 cell surface expression revealed a range of infection efficiencies across the panel of mutants. Our infection results revealed a non-linear relationship between soluble SARS-CoV-2 RBD binding to ACE2 and pseudovirus infection, supporting a major role for binding avidity during entry. While ACE2 mutants D355N, R357A, and R357T abrogated entry by both SARS-CoV and SARS-CoV-2 spike proteins, the Y41A mutant inhibited SARS-CoV entry much more than SARS-CoV-2, suggesting differential utilization of the ACE2 side-chains within the largely overlapping interaction surfaces utilized by the two CoV spike proteins. These effects correlated well with cytopathic effects observed during SARS-CoV-2 replication in ACE2-mutant cells. The panel of ACE2 mutants also revealed altered ACE2 surface dependencies by the N501Y spike variant, including a near-complete utilization of the K353D ACE2 variant, despite decreased infection mediated by the parental SARS-CoV-2 spike. Our results clarify the relationship between ACE2 abundance, binding, and infection, for various SARS-like coronavirus spike proteins and their mutants, and inform our understanding for how changes to ACE2 sequence may correspond with different susceptibilities to infection.

**Data Availability Statement:** All relevant data are within the manuscript and its Supporting Information files.

**Funding:** This research was supported by a National Institutes of Health (NIH) grant AI141620 (KAM). The Cytometry & Imaging Microscopy Shared Resource of the Case Comprehensive Cancer Center was supported by NIH grants P30CA043703 and S10OD021559. The SARS-CoV-2 work was performed in the BSL3 at Case Western Reserve University (CWRU), which is supported by the CWRU and University Hospitals Center for AIDS Research grant P30AI36219. The Genotype-Tissue Expression (GTEx) Project was supported by the Common Fund of the Office of the Director of the National Institutes of Health, and by NCI, NHGRI, NHLBI, NIDA, NIMH, and NINDS. The funders had no role in study design, data collection and analysis, decision to publish, or preparation of the manuscript.

**Competing interests:** The authors have declared that no competing interests exist.

## Author summary

SARS-like coronaviruses, such as SARS-CoV-2, use their spike proteins to bind a common surface on the human ACE2 protein to gain entry and subsequently infect cells. We used site-specific genomic integration and expression of WT ACE2 or its missense variants, many of them previously observed in human exomes, to determine how ACE2 sequence and abundance correspond to infectability by SARS-CoV or SARS-CoV-2. We found that reduced binding only partially corresponded to infection, and mainly only at lower ACE2 abundance levels. We observed some human ACE2 variants differentially affect SARS-CoV, SARS-CoV-2, or SARs-CoV-2 N501Y spike variant pseudovirus entry, showing that each viral spike binds ACE2 in a unique manner. Our results provide improved quantitative understanding for how ACE2 sequence and abundance correlate with infectivity, with implications for how natural human ACE2 variants, or variants observed in related species, may impact susceptibility to infection. These genetic tools can be repurposed to characterize future SARS-CoV-2 spike variants, or to better understand how receptor protein sequences correspond with entry by zoonotic viruses during cross-species transmission events.

## Introduction

Zoonotic spillover of viruses from animal reservoirs can decimate public health systems and the global economy, as evident with the current SARS-CoV-2 pandemic. Successful entry into the cells of another host species is a major step for cross-species virus transmission[1]. Beta coronaviruses of the subgenus Sarbecovirus, also known as Severe Acute Respiratory Syndrome-like coronavirus (SARS-like CoVs), interact with cell-surface ACE2 proteins to enter host cells[2,3].

Similar to MERS-CoV but dissimilar to SARS-CoV, the SARS-CoV-2 spike possesses a furin cleavage site that separates the S1 and S2 units during virus release from producer cells [4]. While this cleavage is not essential and can be performed by host proteases in the target cell, furin cleavage increased pathogenicity in animal models [5,6]. The first essential step for productive SARS-CoV or SARS-CoV-2 infection occurs when the spike protein uses its receptor binding domain (RBD) to interact with cell surface ACE2[2,7]. Upon binding, ACE2-utilizing CoVs undergo a subsequent protease cleavage step to free the fusion peptide, which becomes inserted into the target cell membrane. Cleaved, ACE2-bound spike trimers then undergo a major conformational rearrangement that catalyzes membrane fusion and enables virus entry into the cell cytoplasm. Depending on the cellular conditions, this protease activation step can happen in one of two general ways. Some cells express cell-surface proteases such as TMPRSS2, capable of cleaving the spike protein and enabling virion entry at the plasma membrane[8–11]. In conditions where the necessary cell-surface proteases are absent, endocytosed CoV virions are cleaved and activated by ubiquitous endosomal proteases such as Cathepsin L[12]. The ACE2 binding step is believed to occur similarly in either case, regardless of the second protease-cleavage step that decides the location of viral fusion and entry.

Cell-surface expression of ACE2 is critical for SARS-CoV and SARS-CoV-2 entry, but the exact relationship between the amounts of ACE2 cell-surface expression and the corresponding likelihood of SARS-like CoV entry is not well characterized. The dissociation constant of monomeric human ACE2 ectodomain for the SARS-CoV and SARS-CoV-2 spike RBDs is in the tens of nM range or tighter[13,14]. Importantly, spike—ACE2 interaction during infection is not as isolated monomers, since numerous complexes may form between multiple copies of

cell-surface ACE2 receptors and the two dozen spike trimers found on the surface of the average SARS-CoV-2 virion[15]. When the virus and host membranes are brought into close proximity, local enrichment and avidity-based enhancements may create a feed-forward process after initial contact, using avidity to enhance the likelihood of entry and infection. Thus, monomeric binding, while simple and quantitative, likely hides the true dynamics of spike—ACE2 avidity interactions that occur during virion attachment and entry into cells.

While the importance of ACE2 for SARS-like CoV infection is undisputed, we collectively possess little quantitative understanding of the relationship between ACE2 cell-surface density and successful virus entry. Numerous studies have assessed ACE2 transcripts across tissues and cell types[16], though their implications for actual protein abundance differences are unclear. While immunohistochemical data is more relevant[17], this data is largely observational and not well linked to rates of virus entry or infection. We also lack understanding for how differences in ACE2 protein sequence, such as the missense variants that naturally exist at low frequency in the human population, can affect susceptibilities to infection by SARS-CoV and SARS-CoV-2. Here, we applied a synthetic biology approach to stably overexpress ACE2 or its coding variants at precise levels within HEK 293T cells. We found that high cell-surface ACE2 expression resulted in little difference in pseudovirus entry, even with ACE2 variants known to drastically reduce binding to isolated SARS-CoV or SARS-CoV-2 RBDs. Reducing cell surface ACE2 expression uncovered concomitant reductions in pseudovirus entry from variants known to abrogate binding.

Our panel of over two dozen ACE2 missense variants revealed a non-linear relationship between published changes in monomeric binding and the efficiencies of pseudovirus infection. We found a subset of variants that reduced SARS-CoV or SARS-CoV-2 pseudovirus infection, with two of these variants observed at very low frequencies in human exomes. Importantly, ACE2 variants that reduced SARS-CoV-2 spike pseudovirus infection were well correlated to cytopathic effects observed with SARS-CoV-2 replication. We also observed a distinct pattern of ACE2 variant sensitivities for the SARS-CoV-2 N501Y spike variant as compared to spikes encoding N501 or those encoding D614G, suggesting an alteration in the manner that N501Y spike engages the ACE2 protein surface during infection. Our study provides a convenient cell engineering platform for quantitatively comparing the relationships between receptor sequence and abundance with pseudovirus infection, informing how these factors may converge to affect how efficiently viral particles with SARS-CoV or SARS-CoV-2 spikes enter cells.

## Results

### Achieving pseudo-clonal ACE2 overexpression in human cells

To assist our experiments, we developed an improved version of our previously published Bxb1 landing pad system[18,19]. This system harnesses the high intracellular activity of the Bxb1 bacteriophage recombinase to allow facile genomic integration of exogenous DNA with suitable recombinase recognition sites supplied by transfection. To maximize homogeneity of expression, the system is engineered to encode one Bxb1 attP recombination site behind a genomically integrated Tet-inducible promoter (**Fig 1A**). Upon transfection of a promoterless plasmid encoding the transgene of interest behind an attB recombination site, a codon-optimized Bxb1 recombinase enzyme catalyzes the integration of a single plasmid molecule behind the promoter, enabling single-copy stable transgene expression. By working with a landing pad cell line expanded from a single clone, all of the integrated plasmids enter an identical genomic locus, resulting in pseudo-clonal stable expression of the transgene of interest.

To improve upon previous designs, we developed a version of the landing pad cells that simultaneously encodes both the iCasp9 negative selection gene and the Bxb1 integrase behind

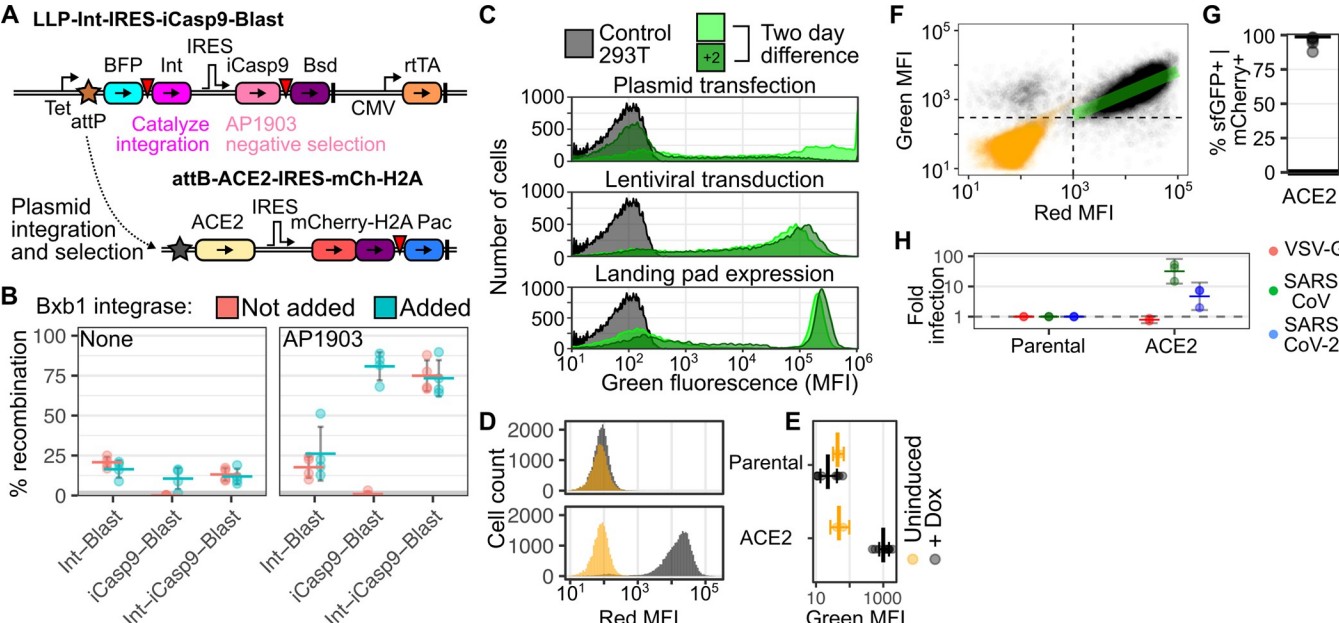

**Fig 1. Improved cell engineering platform expressing human ACE2.** A) Schematic of the newly described LLP-Int-iCasp9-Blast landing pad (top), and the ACE2 expression construct used in the majority of this study (bottom). Key: Tet, Tet-inducible promoter; BFP, Blue fluorescent protein; Int, Bxb1 integrase; IRES, Internal Ribosome Entry Site; iCasp9, inducible caspase 9; Bsd, Blasticidin-S deaminase; CMV, Cytomegalovirus promoter; rtTA, reverse tet transactivator; H2A, Histone 2A; Pac, Puromycin N-acetyltransferase; attP and attB are Bxb1 recombination sites. Inverted red triangles denote viral 2A-like sequences allowing co-translational separation of the polyprotein. Thick vertical lines are transcriptional terminator sequences. B) Percentages of recombined cells in various HEK 293T Bxb1 landing pad cells, in the absence (left) or presence (right) of negative selection agent AP1903. Red and blue indicate the absence and presence of exogenous Bxb1 integrase expression plasmid, respectively. n = 4; error bars denote 95% confidence intervals. C) Representative smoothed histograms of the flow cytometry mean fluorescence intensities of cells where GFP was expressed by plasmid transfection (top), lentiviral transduction (middle), or recombination into the landing pad (bottom). Light and dark green correspond to days 3 and 5 after plasmid transfection, respectively. For the lentivirus transduced and Bxb1 recombined cells, the cells had been generated a week before the first time-point. D) Representative mCherry fluorescence distribution of ACE2-recombined cells, as captured by flow cytometry, in cells left uninduced (orange) or induced to express ACE2 from the Tet-inducible promoter using 2 μM doxycycline (black). E) Geometric means of green fluorescence of SARS-CoV-2 RBD-sfGFP -stained ACE2 expressing cells. n = 2 and 8 for uninduced and induced cells, respectively. Error bars denote 95% confidence intervals. F) Representative scatterplot of mCherry fluorescence and RBD-sfGFP staining of ACE2 recombinant cells. The green line denotes the linear correlation between red and green MFI for the double-positive population. Pearson's $r^2$ = 0.44. G) Percent of mCherry positive cells that were also positive for SARS-CoV-2 RBD-sfGFP staining, for 5 repeat staining experiments. H) Fold pseudovirus infection of ACE2 overexpressing cells, normalized to infection of parental HEK 293T cells. n = 3 for SARS-CoV and SARS-CoV-2 spikes, error bars denote 95% confidence intervals.

the Tet-inducible promoter and a Bxb1 attP recombination sequence (**Fig 1A**). Bxb1 recombinase expressed from the landing pad enables integration of a Bxb1 attB-containing plasmid shortly after transfection, turning off expression of the recombinase enzyme. Plasmid integration also turns off expression of the inducible suicide gene, iCasp9, which dimerizes in the presence of the small molecule AP1903 and triggers apoptosis[20]. Thus, transfection of landing pad cells with a promoterless plasmid followed by negative selection with AP1903 allows for the rapid enrichment of a pseudo-clonal pool of cells homogeneously expressing a single genomically encoded transgene. On average, we found that ~ 12.5% of starting cells could be recombined with a transgene of interest, and this could be rapidly selected to ~ 75% recombined cells upon negative selection after the addition of AP1903 (**Fig 1B**). Flow cytometry of cells transfected with GFP encoding plasmid, cells transduced with GFP-encoding lentiviral vectors, and cells harboring a genomic landing pad with a recombined GFP-encoding attB-containing plasmid, demonstrated the recombined landing pad cells to have optimal characteristics of stable and precise transgene expression compared to the other methods (**Fig 1C**). We thus utilized this Bxb1 landing pad platform for all subsequent cell-engineering experiments.

We exogenously expressed ACE2 using a panel of constructs (**Figs 1D and S1**). Each construct harbored a co-transcriptional or co-translational mCherry reporter gene, thus marking cells modified to express exogenous ACE2 with bright red fluorescence (**Figs 1D, S1A, S1B and S1C**). To assess the relative amounts of ACE2 expressed on the cell surface, we co-incubated the cells with supernatants containing the SARS-CoV-2 spike receptor binding domain (RBD) fused to superfolder GFP[21]. All of the constructs exhibited comparable amounts of overall staining, suggesting roughly equal levels of ACE2 cell-surface expression regardless of the differences in the construct formats (**Figs 1E and S1D**). In contrast, the same cells grown without doxycycline, and thus lacking transgenic ACE2 expression from the Tet-inducible promoter, did not exhibit cell surface ACE2 staining above our limit of detection (**Figs 1E and S1D**). Parental cells lacking transgenically overexpressed ACE2 also did not stain for cell surface ACE2.

We assessed how accurately the mCherry reporter gene marked transgenic cells expressing ACE2, as it is a more convenient and reproducible reporter than cell staining (**Fig 1F**). We found that approximately 95% of the mCherry expressing cells also exhibited cell-surface ACE2 expression (**Figs 1G and S1E**). Even within the double-positive population, we saw that mCherry fluorescence explained 44% of the variance in green fluorescence (Pearson's $r^2$ = 0.44; **Fig 1F, green line**). Thus, we concluded that red fluorescence from mCherry expression could serve as a reliable proxy for identifying ACE2-modified engineered cells in our culture, and we used mCherry positivity to identify ACE2-expressing cells in all future flow cytometry readouts.

## Tuning ACE2 expression for pseudovirus infection

We next measured how much exogenous ACE2 expression from each construct was boosting spike-mediated pseudovirus entry. SARS-CoV and SARS-CoV-2 are both biosafety level 3 pathogens, with SARS-CoV also considered a select agent. Fortunately, pseudovirus models are convenient surrogates for SARS-CoV and SARS-CoV-2 entry, particularly for characterizing the spike interaction with ACE2 during infection[22]. We produced GFP-encoding lentiviral particles pseudotyped by either SARS-CoV spike, SARS-CoV-2 spike, or Vesicular Stomatitis Virus glycoprotein (VSV-G) (**S2A, S2B and S2C Fig**). VSV-G uses LDLR as its entry receptor[23], thus serving as a negative control for ACE2-dependent entry in our assay. When these particles were added to cells expressing each construct, we observed roughly 10- to 100-fold increased entry and infection relative to unmodified parental cells for the two coronavirus spikes (**Figs 1H and S2D**). In contrast, VSV-G pseudoviruses did not exhibit any enhancement. We proceeded with the ACE2 expression vector cotranscriptionally encoding histone-fused mCherry for all subsequent experiments (**Figs 1A and S1A**), as this construct performed well in both flow cytometry and microscopy readouts.

We next examined how alterations to the sequence of ACE2 affects its interaction with SARS-CoV and SARS-CoV-2 spike. Cocrystal structures between ACE2 and SARS-CoV-2 spike RBD show a clear set of ACE2 residues involved in the interaction, including K31 and K353[7,24,25] (**Fig 2A**). Accordingly, mutation of the ACE2 side-chains at the interface can disrupt the interaction. For example, the K31D and K353D ACE2 variants are known to abrogate interaction with soluble monomeric RBDs from SARS-CoV[26] and SARS-CoV-2[21]. We thus engineered cells overexpressing ACE2 K31D or K353D and exposed them to SARS-CoV or SARS-CoV-2 pseudoviruses, including a construct where the entire ACE2 ectodomain was deleted (dEcto) to serve as a negative control (**Fig 2B**). We found that cells expressing the K31D or K353D ACE2 variants reduced cell surface staining with SARS-COV-2 RBD-sfGFP by 42- and 73-fold, respectively, to levels at or near the background of the assay (**Fig 2C**). In

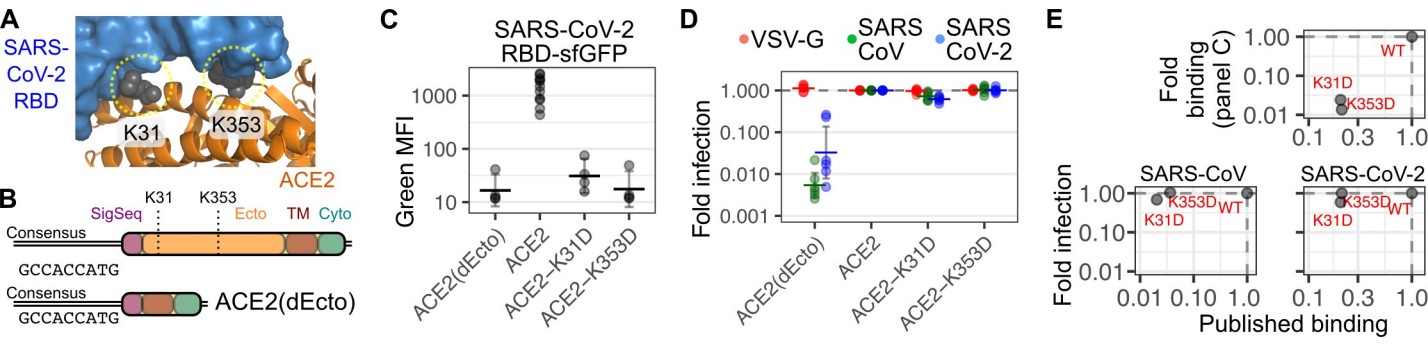

**Fig 2. ACE2 variant expressed at high abundance levels.** A) The location of ACE2 (orange) K31 and K353 residues (grey) in the interface with SARS-CoV-2 RBD (blue); with the three-dimensional structure provided by PDB 6M17. B) Schematic showing the coding region of the ACE2 expression construct, as well as the dEcto negative control construct lacking the entire extracellular domain. C) ACE2 variant constructs encoding the consensus Kozak, stained with SARS-CoV-2 RBD-sfGFP. n = 4 or more, error bars denote 95% confidence intervals. D) Pseudovirus infection rates of ACE2 variants encoding the consensus Kozak, normalized to cells encoding WT ACE2. n = 6, error bars denote 95% confidence intervals. E) Scatterplots depicting pseudovirus infection or SARS-CoV-2 RBD-sfGFP staining of cells encoding consensus Kozak ACE2 preceding WT, K31D, or K353D ACE2, compared to published binding studies with SARS-CoV or SARS-CoV-2 RBDs.

contrast, these same cells overexpressing ACE2 K31D or K353D were nearly as susceptible to infection by SARS-CoV or SARS-CoV-2 pseudoviruses as cells overexpressing WT ACE2, with the K31D ACE2 variant reducing SARS-CoV and SARS-CoV-2 infection 1.5- and 1.7-fold, respectively, while K353D had no effect on infection with either spike (**Fig 2D**). The similarity of our monomeric cell-surface binding measurements to previously published binding results[21,26], but sharp contrast with our measured rates of infection, suggested a disparity in the phenotypic effects captured by these two assays (**Fig 2E**).

We hypothesized that the high levels of ACE2 expression from the Tet-inducible promoter combined with avidity effects at the cell surface were compensating for the loss of affinity from ACE2 mutation. To test this, we reduced the amount of ACE2 cell surface abundance of our system by altering the Kozak sequence preceding the ACE2 open reading frame, thus reducing its rate of translation[27] (**Fig 3A**). We found that a suboptimal Kozak sequence of "CATTGT", when placed in front of EGFP, reduced green fluorescence over 30-fold (**S2E Fig**). We thus determined whether a suboptimal Kozak sequence could alter ACE2 expression and thus impact SARS-CoV-2 RBD binding. Cells encoding ACE2 behind this suboptimal Kozak sequence exhibited a background level of fluorescence when stained with SARS-CoV-2 RBD-sfGFP (**Fig 3A, bottom**), suggesting that we were reaching the limit of detection of that method. Cells encoding this suboptimal Kozak sequence reduced ACE2 expression roughly 50- to 60-fold by Western blot (**Figs 3B and S3A**), and repeated cell surface ACE2 staining with SARS-CoV-2 RBD-sfGFP was reduced at least 56-fold (**Fig 3C, left**). Notably, the levels of SARS-CoV or SARS-CoV-2 spike-mediated pseudovirus entry were diminished approximately 9.8- and 6.9-fold, respectively, while entry with VSV-G was unaffected (**Fig 3D, left**), suggesting that reducing ACE2 expression can limit the efficiency of spike-mediated viral entry.

We next determined whether the K31D and K353D ACE2 variants could impact the efficiencies of pseudovirus entry when tested under these limited abundance levels. We were unable to measure any further reductions to cell-surface binding of SARS-CoV-2 RBD-sfGFP due to the reduced expression from the suboptimal Kozak (**Fig 3C, right**). Nevertheless, we found that both ACE2 variants reduced spike-mediated infection for both pseudoviruses. SARS-CoV was reduced 3.2-fold by K31D and 6.8-fold by K353D, while SARS-CoV-2 was reduced 5.2-fold and 6.4-fold (**Fig 3D, right**). This supported our interpretation that reduced ACE2 cell-surface expression was needed to reveal how alterations to the ACE2 sequence could impact infection.

To better contextualize the ACE2-dependent infection we observed with the suboptimal Kozak ACE2 construct, we compared ACE2 abundance levels between unmodified HEK 293T

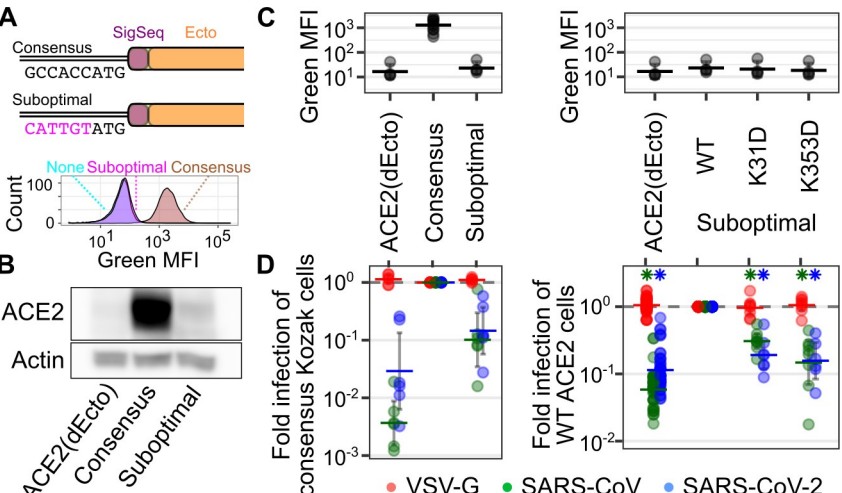

**Fig 3. Tuning of ACE2 cellular abundance using suboptimal Kozak sequences.** A) (Top) Schematic of consensus and suboptimal Kozak sequences preceding the ACE2 coding region. (Bottom) Representative flow cytometry smoothed histograms of negative control cells, or cells expressing WT ACE2 encoded behind a consensus or suboptimal Kozak sequence, stained with SARS-CoV-2 RBD-sfGFP. B) Representative Western blot showing the differences in total ACE2 protein abundance in engineered cells. C) Cell surface staining using SARS-CoV-2 RBD-sfGFP for cells with contrasting Kozak sequences (left), or encoding ACE2 variants (right). n = 4 or more, error bars denote 95% confidence intervals. D) Pseudovirus infection of cells with contrasting Kozak sequences (left), or encoding ACE2 variants in the context of a suboptimal Kozak (right). n = 6 or more, error bars denote 95% confidence intervals. Asterisks denote samples that exhibited a p-value less than 0.01 in a one-sample T-test from the infection value of 1.

cells, which are generally considered non-permissive to SARS-CoV-2 entry, with Vero E6 cells, which are widely considered permissive to SARS-CoV-2 infection, and are commonly used to propagate the virus in BSL3 laboratories [28]. We found that these suboptimal Kozak ACE2 cells exhibited roughly 4-fold more ACE2 protein than unmodified HEK 293T cells (**S3B and S3C Fig**). In contrast, the suboptimal Kozak ACE2 cells exhibited roughly 4-fold less total ACE2 protein than Vero E6 cells. With these approximate values in mind, we queried the Genotype-Tissue Expression (GTEx) project portal to estimate which cell types and tissues have ACE2 transcript levels that may render them similar to the level of ACE2-dependent SARS-CoV-2 spike-mediated entry as assessed with our cell models. HEK 293T cells have a small but non-zero value of 0.1 ACE2 transcripts per million (TPM) within the GTEx database (**S3D Fig**). Projecting a 4-fold increase in ACE2 expression, equivalent to the 4-fold relative increase in ACE2 abundance in the suboptimal Kozak cells, revealed 53% of assessed tissues and cell types had equal or greater ACE2 transcripts. Projecting a 16-fold increase, approximating a relative increase to ACE2 protein abundance seen in Vero E6 cells, revealed 25% of assessed tissues and cell types with equal or greater transcripts. Lung tissue exhibits 0.8 ACE2 TPM, partway between the estimated corresponding levels for suboptimal Kozak and Vero E6 cells (**S3D Fig**). While these are rough estimates and should be approached with caution, they suggest that a sizable fraction of human tissues and cell types may express enough ACE2 to permit at least low-level ACE2-dependent entry seen with our suboptimal Kozak cells.

## Examining ACE2 variant effects during infection

With our assay better calibrated, we next probed these interactions with additional ACE2 variants at the interaction interface, all in the context of reduced expression from the suboptimal Kozak. This included Q42R, Y83F and E329K, predicted to disrupt hydrogen bonds with

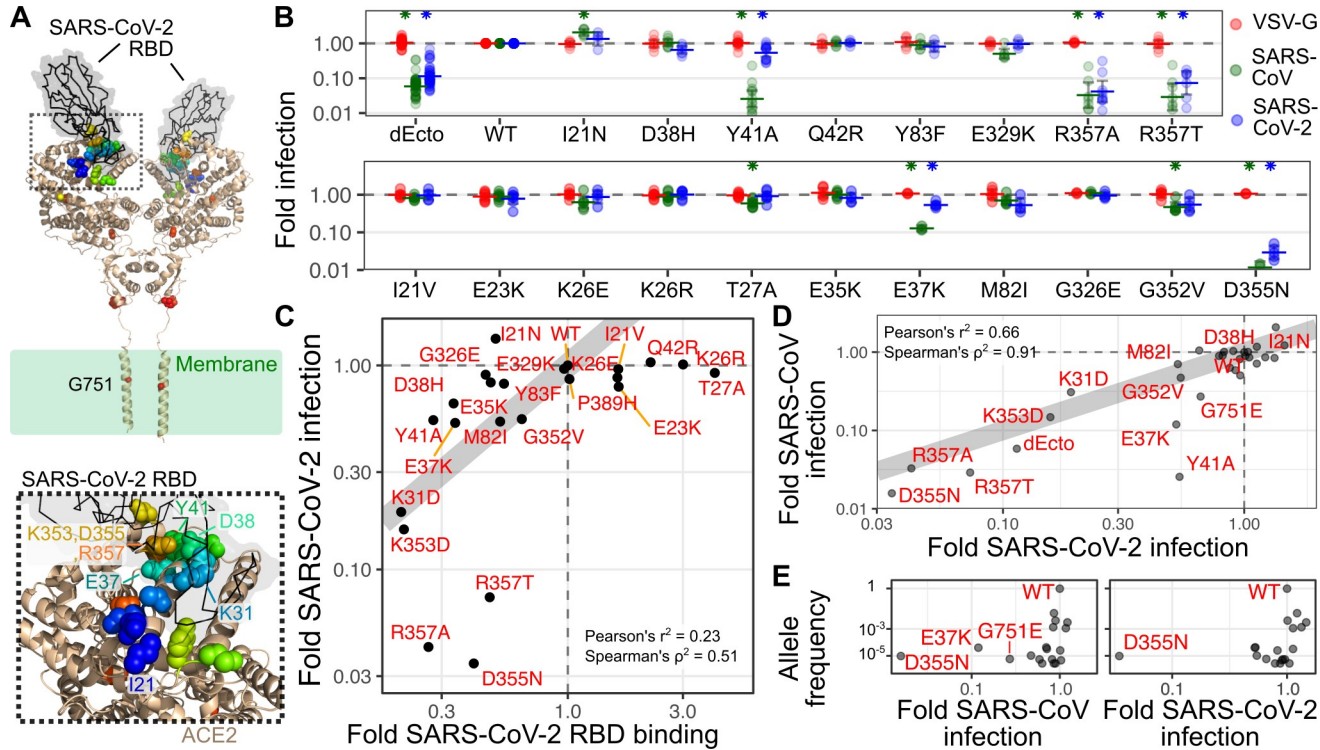

**Fig 4. ACE2 variant effects on binding and infection.** A) Structure of the SARS-CoV-2 RBD in complex with ACE2, from PDB 6M17. Positions of variants tested in the manuscript are shown as spheres colored according to their order from the N- to C-terminus. (Bottom) A zoomed in view of the RBD:ACE2 interface, where the majority of the tested variants are found. B) Infection of a panel of ACE2 variants of residues near the spike interaction surface, separated into variants currently unobserved (top) and observed (bottom) in publicly available human genomics datasets. n = 5 or more replicates, error bars denote 95% confidence intervals. C) Scatter plot showing relative pseudovirus infection rates (y-axis) and the mean value of the published SARS-CoV-2 RBD-sfGFP staining (x-axis) for WT ACE2 and its variants. D) Scatter plot showing normalized infection of SARS-CoV-2 (x-axis) or SARS-CoV (y-axis) spike pseudoviruses. E) Scatter plot comparing the pseudovirus infection values of SARS-CoV or SARS-CoV-2 pseudoviruses with their average allele frequencies observed in the GnomAD and BRAVO exome databases. The grey line in panels C and D, denote a hypothetical perfect correspondence between assays with a slope of 1.

SARS-CoV RBD, and D38H and Y41A predicted to disrupt hydrogen bonds seen with both RBDs. We included three additional variants, I21N, R357A, and R357T, all near the interaction interface (**Fig 4A, bottom**). Of these variants, R357A and R357T had the strongest overall effect, reducing SARS-CoV pseudovirus infection 30.5-fold and 34.5-fold while reducing SARS-CoV-2 24-fold and 13.7-fold, respectively. Y41A had a strong effect on SARS-CoV pseudovirus infection, reducing it 39.1-fold while only affecting SARS-CoV-2 1.9-fold (**Fig 4B, top**). Interestingly, I21N significantly increased SARS-CoV pseudovirus infection approximately 2.1-fold (**Fig 4B, top**).

We also tested a panel of 18 variants observed in humans (**Figs 4B and S4A**), to determine if any rare germline variants could impact spike-mediated viral entry. There are 276 unique ACE2 missense variant alleles listed in GnomAD and BRAVO[29]. We first tested a panel of seven variants (G211R, P389H, T519I, S692P, N720D, L731F, and G751E) located across the entire ACE2 protein (**S4A Fig**). While found outside of the SARS-CoV and SARS-CoV-2 spike interaction interface, each of these variants had Combined Annotation Dependent Depletion (CADD) scores ranging from 14.2 to 25.8, suggesting they may be disruptive for normal ACE2 folding and function[30]. All of these ACE2 alleles promoted SARS-CoV and SARS-CoV-2 pseudovirus infection, with the 3.7-fold decrease to SARS-CoV pseudovirus infection observed with G751E as the only exception. G751 is found in the transmembrane domain of ACE2

(**Fig 4A**), and substitution of transmembrane residues to charged amino acids such as glutamate are generally poorly tolerated[31]. Consistent with this, Western blotting analysis revealed the total abundance of G751E to be moderately reduced relative to WT (**S4B Fig**).

We next focused on eleven human variants of residues close to the CoV spike interface: I21V, E23K, K26E, K26R, T27A, E35K, E27K, M82I, G352V, G362E, and D355N (**Fig 4B, bottom**). T27A and G352V significantly reduced SARS-CoV entry, but the effects were 2.1-fold or less. In contrast, E37K significantly reduced entry by both viruses, though it had a disproportionately larger 8.4-fold reduction to SARS-CoV spike-mediated infection, rather than the 1.8-fold reduction observed with SARS-CoV-2. D355N completely abrogated ACE2-dependent infection by both pseudovirus spike proteins (**Fig 4B, bottom**).

While the effects of these variants on pseudovirus infection were clear, we wished to further distinguish whether these effects were due to altered protein abundance or binding. We thus performed Western blotting for lysates from eight additional variants spanning the range of observed pseudovirus infectivities (**S4B** and **S5** **Figs**). Across the replicates, total abundance of WT ACE2 translated behind the suboptimal Kozak was only ~ 2.5-fold increased over the amount of endogenous protein abundance already present in 293T cells (**S4B Fig**), and these ACE2 variants spanned abundances within this range. Aside from the reduced ACE2 abundance of G751E encoding cells, the R357A variant was the only other variant with clearly reduced abundance, with overall protein levels similar to the dEcto control. While the E37K, Y41A, K353D, and D355N had slightly variable abundance depending on the replicate, the protein levels were still on average in the range of the WT ACE2 protein.

To better delineate the variants affecting pseudovirus infection by altered abundance or altered binding, we created a scatter plot comparing ACE2 protein abundance with SARS-CoV or SARS-CoV-2 pseudovirus infection (**S4C Fig**). The reduced pseudovirus infection observed with R357A and G751E appears largely due to reductions in protein abundance. In contrast, K353D, D355N, and R357T each permitted less infection than would be expected based on their effects on protein abundance alone, suggesting their effects indeed due to altered binding. While the slight effect of E37K and Y41A on SARS-CoV-2 may be due to their reduced protein abundance, their disproportionately large reduction to SARS-CoV pseudovirus infection suggests an important role in altering the binding interface with SARS-CoV RBD. Altogether, our results show that reduced pseudovirus infectivity observed with ACE2 variants can be due to a combination of altered total protein abundance and physical disruption to the binding interface.

We compared our full dataset to the aforementioned studies examining how mutation of human ACE2 can affect binding to soluble, monomeric SARS-CoV or SARS-CoV-2 RBD *in vitro*. We observed a clear concordance (n = 5, Pearson's $r^2$ = 0.83, p = 0.03) between SARS-CoV RBD binding and infection with SARS-CoV pseudoviruses, though the K31D and K353D variants promoted infection better than expected based on binding, while the Y41A variant promoted infection worse than expected (**S4D Fig**).

We expanded our comparison to look at correlations between ACE2 variants and SARS-CoV-2 spike binding and infection. A recent deep mutational scan of the interface residues of ACE2 yielded ~2,000 variants that were scored for their abilities to bind soluble SARS-CoV-2 spike RBD[21]. While the primary goal of this study was to find ACE2 variants with enhanced RBD binding that could be used for engineered biological therapeutics for SARS-CoV-2 infection, the dataset also revealed a large number of ACE2 variants with reduced SARS-CoV-2 spike RBD binding in this format. Thus, we determined whether this dataset can also inform our understanding for how a range of ACE2 sequence variants differentially bind the SARS-CoV-2 RBD to alter the efficiency of SARS-CoV-2 spike-mediated infection.

We observed a correlated but non-linear relationship between the ACE2 variant RBD binding values and pseudovirus infection (n = 23, Pearson's $r^2$ = 0.23, Spearman's $\rho^2$ = 0.51;

**Figs 4C** and **S4E**). The greatest discrepancies were observed at the extreme ends of the binding spectrum. ACE2 variants I21V, E23K, K26E, K26R, T27A, and Q42R all exhibited increased binding for soluble SARS-CoV-2 RBD, but SARS-CoV-2 spike pseudoviruses did not enter cells expressing these variants any more frequently than cells expressing WT ACE2. Furthermore, some variants that were 2 to 4-fold reduced in binding exhibited anywhere between 0.5 to 30-fold decreases to infection. This included I21N, which bound the SARS-CoV-2 RBD ~ 2-fold worse than WT when in solution, and yet permitted 1.4-fold increased infection relative to WT for both SARS-CoV and SARS-CoV-2 spikes (**Fig 4B**). In contrast, D355N, R357A, and R357T exhibited similar ~ 2.5-fold average reductions in binding, and yet permitted an average ~ 29-fold reduction in pseudovirus infection. This data helps calibrate existing ACE2—spike binding data, so that its limitations for extrapolation to infection can be better understood and predicted.

## Differential ACE2 variant reliances during CoV infection

The complete panel of 30 total ACE2 variants tested for both SARS-CoV and SARS-CoV-2 pseudovirus infection allowed us to compare how each spike was interfacing with the ACE2 protein surface during infection. Overall, there was a high, linear correlation between SARS-CoV and SARS-CoV-2 pseudovirus entry across the variants (n = 30, Pearson's $r^2$ = 0.66, p = 6 x $10^{-8}$). The clear exceptions to the correlation were Y41A and E37K, which both disproportionately reduced SARS-CoV pseudovirus infection while exhibiting little effect on SARS-CoV-2 (**Fig 4D**).

We next assessed how frequently the ACE2 variants that diminish SARS-CoV and SARS-CoV-2 spike mediated infection are found in the human population (**Fig 4E**). ACE2 is found on the X chromosome. It's expression in females is seemingly complex, as it is encoded on a region that escapes X chromosome inactivation, but its level of expression is often tissue-specific, with some tissues exhibiting lower expression in females as compared to males[32]. Thus, it is unclear whether females heterozygous for an ACE2 variant allele may possess cells that express the variant allele in the absence of large amounts of the WT allele. On the other hand, males that receive a variant ACE2 allele from their mother would be hemizygous for ACE2, thus only capable of expressing the variant allele. The D355N variant, which abrogated pseudovirus infection mediated by both spikes, has an observed allele frequency of 7.6 x $10^{-6}$ in GnomAD, and no hemizygous individuals were observed. E37K, which selectively diminishes SARS-CoV pseudovirus without affecting SARS-CoV-2 pseudovirus infection, is more frequently observed with an allele frequency of 5.4 x $10^{-5}$ (**Fig 4E**). Notably, there were 7 hemizygous individuals observed between the GnomAD and BRAVO variant databases. Thus, while rare, there are likely a small number of individuals that currently exist in the world population that harbor ACE2 alleles that may reduce how well SARS-CoV or SARS-CoV-2 enter their cells.

We also assessed whether any of these infection-disrupting variants were observed in other species. Neither D355N, R357A, nor R357T variants were observed in other sequenced species. In contrast, K31 is positively selected in bats[33], with multiple *Rhinolophus* species such as *R. ferrumequinum*, *R. landeri*, *and R. pusillus* encoding D31 (**S6 Fig**). Notably, Glu, Asn, and Thr residues are also observed at position 31 in other bat species, including high intraspecies variation within *R. sinicus* (**S6 Fig**). Despite the importance of K31 for SARS-CoV-2 infection of cells expressing human ACE2, its effect in the context of the bat ACE2 alleles has not been characterized, and it remains to be seen whether these residues are affecting SARS-CoV-2 infection within bat cells. Notably, the conversion of murine ACE2 to N31K and H353K renders *Mus musculus*, house mouse cells permissible for SARS-CoV-2 entry, supporting the critical role of variants at this position across species[34].

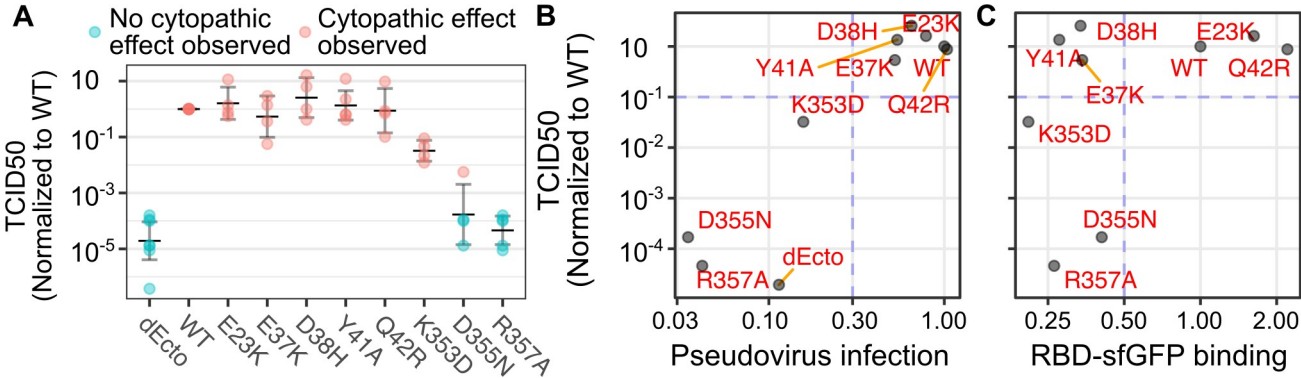

**Fig 5. Comparison of ACE2 variant susceptibilities to SARS-CoV-2 replication with SARS-CoV-2 spike pseudovirus infection.** A) Tissue culture infectious dose 50 assays, where dilutions of SARS-CoV-2 were added to 96-wells containing HEK 293T cells expressing the denoted WT or variant ACE2 proteins. TCID50 values were normalized to values corresponding to infection of WT ACE2 cells assessed in the same experiment. Samples colored in blue are those where no cytotoxicity was observed in any dilution, and these samples were given a nominal count of 1 to denote the limit of detection of the assay, and were then normalized accordingly. The horizontal dashes indicate geometric means across experiments. Error bars denote 95% confidence intervals. N = 3 or more replicates. B) Scatter plot comparing SARS-CoV-2 pseudovirus infection and geometric means of normalized TCID50 values shown in panel A. Blue lines indicate representative threshold values that separate permissive from non-permissive ACE2 variants. C) Scatter plot comparing published SARS-CoV-2 RBD-sfGFP binding values to geometric means of normalized TCID50 values.

To assess whether the ACE2 variant reliances measured with SARS-CoV-2 pseudoviruses correlated to infection with actual, replicating SARS-CoV-2, we exposed a subset of ACE2 variant cells to the WA1 SARS-CoV-2 isolate. We observed that 293T cells expressing WT ACE2 exhibited clear cytopathic effects that spread throughout the entire well, while cells expressing the ectodomain-deleted allele (dEcto) were completely protected from virus induced cytopathy. To quantitate the effects of ACE2 variants on virus spread, we performed tissue culture infectious dose 50 (TCID50) assays, which can be used to calculate the relative titer of the virus on cells with variant ACE2 proteins (**Fig 5A**). We observed a strong correspondence between infection with SARS-CoV-2 pseudovirus and SARS-CoV-2 titer calculated by TCID50 (**Fig 5B**). The ACE2 variants that supported less than 30% of WT ACE2 pseudovirus infection exhibited less than 10% of cytopathic effect quantitated by TCID50, with R357A incapable of exhibiting any cytopathic effect. Conversely, the ACE2 variants that supported more than 30% of WT ACE2 pseudovirus infection also supported more than 10% of cytopathic effect quantitated by TCID50. Finally, we compared the multi-cycle SARS-CoV-2 replication results to the deep mutational scanning SARS-CoV-2 RBD-sfGFP binding data (**Fig 5C**). We saw a similar relationship as before, where some variants that no longer bound SARS-CoV-2 RBD-sfGFP still permitting SARS-CoV-2 replication to WT levels. Thus, ACE2 variants could disrupt SARS-CoV-2 spread, and pseudovirus infection assays could serve as a reliable proxy to characterize these effects.

The panel of ACE2 variants we generated also provided a unique means for characterizing SARS-CoV-2 spike variants. Numerous spike variants have emerged over the course of the pandemic. This includes D614G, which emerged following initial spread into Europe[35], and the N501Y variant, which more recently emerged within the B.1.1.7, B.1.351, and P.1 lineages. D614 is found outside of the receptor binding domain, and is thought to increase spike protein density on viral particles and to increase their infectivity[36]. In contrast, N501Y is found on the surface of the RBD that directly contacts ACE2. While it confers a slight advantage for antibody escape[37], it also slightly enhances binding with human ACE2[38] and drastically enhances binding to murine ACE2[39]. New assays are needed to isolate each potential function, so that the potential multifaceted roles of each spike variant can be separated and individually characterized.

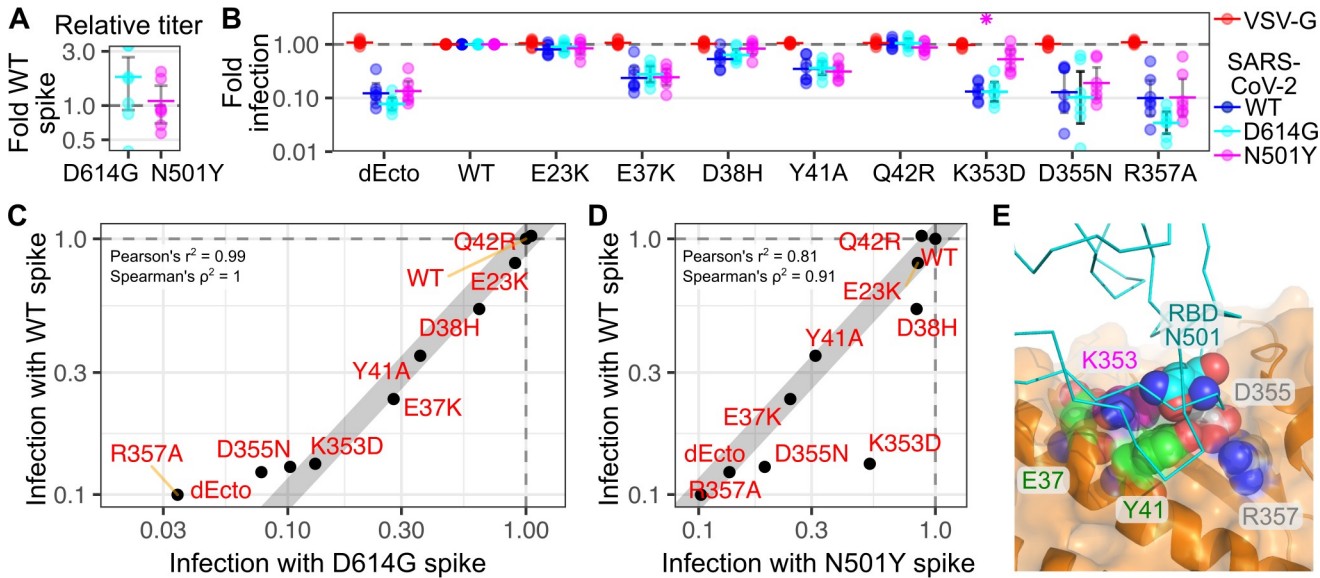

**Fig 6. ACE2 variant dependencies for SARS-CoV-2 spike variants.** A) Relative titers of D614G and N501Y SARS-CoV-2 spike pseudoviruses, relative to WT. n = 7. B) Relative infectivity of WT, D614G, N501Y SARS-CoV-2 spike pseudoviruses, normalized to infectivity in WT ACE2 cells. VSV-G pseudoviruses were included as a negative control. n = 7. C) Scatter plot showing normalized infectivities of D614G (x-axis) and WT (y-axis) SARS-CoV-2 spike pseudoviruses. D) Scatter plot showing normalized infectivity of N501Y (x-axis) and WT (y-axis) SARS-CoV-2 spike pseudoviruses. The grey line in panels C and D, denote a hypothetical perfect correspondence between assays with a slope of 1. E) Structure of the RBD:ACE2 interface (pdb: 6m17), with the RBD shown as a cyan ribbon, and ACE shown as an orange cartoon with a semi-transparent surface representation. Residues of ACE variants highlighted in Figs 4E and 6D are shown as sphere representations. N501 on the RBD is shown as blue spheres.

We thus used the ACE2 variant panel to further assess how the N501Y spike variant may differently utilize the ACE2 surface. Similar to previous observation[36], D614G spike pseudoviruses exhibited slightly increased infectivity on cells with WT human ACE2 (**Fig 6A**). In contrast, overall infectivity by N501Y appeared comparable to the original Wuhan WT sequence (**Fig 6A**). We next exposed a panel of ACE2 variants to pseudoviruses coated with the SARS-CoV-2 spike variants, and assessed whether either spike variant had altered infectivity compared to the WT spike (**Fig 6B**). Within the panel, only N501Y infection of K353D ACE2 variant cells exhibited a significant difference, as N501Y was only inhibited 1.9-fold by the K353D variant, while the WT or D614G spike pseudoviruses were inhibited 7.6-fold and 7.7-fold, respectively (**Fig 6B**).

We next assessed whether the panel of ACE2 variants as a whole could reveal larger patterns differentiating how the spike variants engaged ACE2. We found that infection with D614G spike pseudoviruses correlated perfectly with infection by WT spike, with the only loss of linearity observed at the lowest levels of infection (Pearson $r^2$ = 0.99, Spearman $\rho^2$ = 1; **Fig 6C, left**). In contrast, there was a clear difference in correlation between infection with N501Y and WT spike pseudoviruses (Pearson $r^2$ = 0.81, Spearman $\rho^2$ = 0.91; **Fig 6C, right**). While most variants still exhibited a near perfect linear relationship, variants D38H, D355N, and K353D deviated from this relationship. All of these deviations were cases where the N501Y variant was less sensitive to ACE2 mutation than the WT spike. Notably, the pattern of differential ACE2 variant sensitivity for N501Y was distinct from that of SARS-CoV, when compared to SARS-CoV-2 (**Figs 4D and 6D**). Thus, ACE2 variants can uncover differential patterns of ACE2 surface utilization, not only by different viruses such as SARS-CoV and SARS-CoV-2, but even for individual variant spikes encoded by SARS-CoV-2.

Altogether, we developed an improved cell engineering system enabling the facile yet precise genetic interrogation of host proteins that permit viral entry, and demonstrate its utility

for understanding how ACE2 sequence and expression level correspond with infection by SARS-CoV and SARS-CoV-2 spike pseudoviruses. We furthermore demonstrated that ACE2 variants that reduced pseudovirus infection also reduced SARS-CoV-2 replication, and the panel of ACE2 variants can identify SARS-like CoV spike proteins that engage human ACE2 in distinct manners. Our work clarifies our understanding of how SARS-like CoV spike proteins engage human ACE2 during infection, and provides an experimental template for performing similar studies of virus-host interactions.

## Discussion

The interaction between cell surface ACE2 and the spike proteins from SARS-like coronaviruses is a key step during its infection process, and a major determinant for cross-species transmission. For example, mismatched protein sequences between murine ACE2 and SARS-CoV-2 prevents use of traditional mouse models for studies of SARS-CoV-2 pathogenesis, necessitating studies with human ACE2 transgenic mice[40,41], or a mouse-adapted form of SARS-CoV-2 spike[39,42]. We established a synthetic biology approach capable of precisely interrogating the contributions of important residues within key host factors like ACE2, specifically during viral entry. To achieve this, we improved upon our previous cell engineering designs to create a HEK 293T cell line that can easily and rapidly generate stable cells genetically engineered to express human ACE2 or its variants. Applying this system to study SARS-like CoV spike interactions with human ACE2 and its variants, we characterized how two variables of ACE2, protein abundance and sequence, each impact pseudovirus infection. While we did not test TMPRSS2 co-expression in our experiments, the ACE2 binding step is thought to be an independent upstream event, so the ACE2 variant results will likely be similar regardless of whether the virus is subsequently cleaved at the cell surface by TMPRSS2 or endosomally cleaved by Cathepsins.

We tested two contrasting levels of ACE2 expression to assess how ACE2 cell surface abundance correlated with pseudovirus infection. We achieved high ACE2 expression using a Tet-inducible promoter paired with a consensus Kozak sequence preceding the ACE2 coding region. This level of expression is likely comparable to that achieved by more traditional approaches expressing ACE2 via transient transfection, which also exhibits strong staining by SARS-CoV-2 RBD-sfGFP[21]. This expression level was incompatible for studying the impacts of ACE2 coding variants on infection, as there were little to no differences in pseudovirus infection rates with known loss-of-binding ACE2 variants[21,26]. We thus reduced ACE2 abundance roughly 50-fold by replacing the consensus Kozak sequence with a known suboptimal sequence, "CATTGT[ATG]"[27] (**Fig 3B**). Notably, this reduction in protein abundance caused an approximate 7- to 10-fold reduction to infection with WT ACE2 (**Fig 3D**), suggesting this lowered abundance level was limiting for pseudovirus infection. The lowered ACE2 abundance translated from the suboptimal Kozak resulted in only ~ 3 to 4-fold increased abundance over endogenous ACE2 expressed in HEK 293T cells (**S3C Fig**), though this conferred a 10-fold or greater enhancement to infection over the dEcto cells (**Fig 3D**). This non-linearity between ACE2 abundance and pseudovirus infection suggests that the high level of ACE2 translation with the consensus Kozak is reaching saturating levels for infection, while the low levels normally present in HEK 293T cells are severely limiting for infection.

Using endogenous ACE2 expression from Vero E6 cells as an additional reference, we found that the suboptimal Kozak ACE2 cells exhibited ~ 4-fold less ACE2 abundance, while ACE2 protein in consensus Kozak cells harbored vastly more. Extrapolating these relative values to tissues and primary cell types assessed in GTEx showed that a sizable fraction of samples had projected ACE2 expression values comparable to our suboptimal Kozak ACE2 HEK 293T cells. Importantly, this rough estimate is only for ACE2-dependency during entry, and other

factors such as availability of Cathepsin or TMPRSS2 proteases are unaccounted for. Furthermore, this estimate does not factor in replication within cells, or the effects of immune signaling and response during infection. Regardless, this estimate may better contextualize the range of cells and tissues that may become infected, and help steer future studies.

Further studies are needed to better quantify the relationship between ACE2 cell surface abundance and the efficiency of SARS-CoV-2 spike-mediated infection. These results also contextualize the ACE2 abundance levels that are likely achieved through more traditional ACE2 expression via lentiviral transduction, where each cell within the polyclonal population is exhibiting a singular level of permissivity to SARS-CoV-2 spike pseudovirus entry within a wide range present in the overall population. This heterogeneity may confound experiments, since polyclonal ACE2 transduced cells, while in one sense considered isogenic, will likely be phenotypically quite variable for infection, which may affect single-cell assays like CRISPR screens[43–46]. Recent studies have revealed differing efficiencies of antibody neutralization depending on the amount of ACE2 on target cells. For example, antibodies S309 and S2X333 exhibited impaired neutralization of SARS-CoV-2 spike pseudoviruses in ACE2 overexpressing 293T cells as compared to other cell lines such as Vero E6 cells[47]. Thus, understanding and precise control of ACE2 abundance in model systems may be critical for accurate measurement of biological phenomena surrounding spike-mediated entry.

We did not observe significant effects of ACE2 variants K31D or K353D on pseudovirus infection when translated behind the consensus Kozak sequence, consistent with the seemingly saturating levels of ACE2 for pseudovirus infection at high expression levels. Even when highly abundant, these variants exhibited reduced binding with isolated, soluble RBD, either as a fluorescent cell-surface stain[21] or as a more traditional pull-down assay[26], and we recapitulated this observation with SARS-CoV-2 RBD-sfGFP staining (**Fig 2C**). The same cells exhibited no difference to infection when exposed to either SARS-CoV or SARS-CoV-2 spike coated pseudoviruses. This suggests that binding avidity, particularly between proteins located on opposing plasma membranes, has a major influence on pseudovirus infection. While binding assays with isolated RBDs are convenient and informative, the binding results should not be directly extrapolated to interpret efficiencies of infection, due to this discrepancy. As cell-surface ACE2 abundances on the key target cells during infection are not yet precisely characterized, it is currently unclear what ranges of ACE2 cell surface expression are most physiologically relevant. Conversely, until absolute measurements of ACE2 cell surface abundances are precisely captured and compared across primary cells and *in vitro* model systems, researchers should avoid overinterpretation of results obtained with overexpression constructs.

Over the pandemic, numerous manuscripts have speculated how individuals harboring ACE2 variants may have altered susceptibilities to SARS-CoV-2 infection without providing any functional characterization, and the actual impacts to infection have been unclear. Since ACE2 is encoded on the X-chromosome, hemizygous males with ACE2 variants that reduce SARS-CoV-2 spike interaction may be protected from infection. Due to X-inactivation, females carrying one such allele may also exhibit partial protection, though this effect may be tissue specific[32]. Notably, likely due to the critical role of ACE2 in the renin-angiotensin system, only about 9.7% of the expected number of nonsense, splice acceptor, or splice donor variants have been observed in GnomAD v2.1.1[29], suggesting population-level selection against ACE2 loss-of-function variants, including those that reduce ACE2 steady-state abundance. Thus, ACE2 variants that drastically reduce ACE2 folding or subcellular trafficking are likely to be infrequently observed in the population. This is consistent with our observation that none of the human variants exhibiting drastically reduced ACE2 abundance was observed at allele frequencies greater than $1 \times 10^{-5}$.

In contrast, ACE2 variants at the SARS-CoV-2 spike interface, most of which would not be expected to affect its role in the renin-angiotensin system, will not necessarily be under the

same selection pressure, as their roles in regulating the renin-angiotensin system would largely remain undisturbed. There may even be a survival advantage to protein sequence changes that disrupt infection by SARS-like CoVs over evolutionary history. For example, ACE2 K31 is positively selected in bats[33], and may be reflective of ongoing conflict with ACE2 utilizing coronaviruses. Two human variants, E37K and D355N, both reduced SARS-CoV spike mediated infection, with D355N also abrogating SARS-CoV-2 pseudovirus infection in our assay. E37K was also observed in multiple hemizygous individuals, which may render those individuals resistant to SARS-CoV entry. Both of these alleles are rare in the population, and are thus likely to have a very limited effect on epidemic or pandemic spread of the virus. We only tested 18 of the 276 unique ACE2 missense variant alleles currently listed in GnomAD and BRAVO, or of the 4,749 total missense ACE2 variants possible through single nucleotide variation across its full coding region. Thus, while our study is far from comprehensive in studying potential impacts of human ACE2 missense variants on SARS-CoV or SARS-CoV-2 spike mediated entry, the collective rarity of ACE2 missense germline variation in humans suggests that they will likely have only limited effect on SARS-CoV-2 entry and infection at the population level.

We found that different spike proteins were differentially sensitive to the panel of ACE2 variants that we tested. Overall, SARS-CoV spike pseudoviruses were sensitive to more ACE2 sequence variants in the panel than those with SARS-CoV-2 spikes. In contrast, the N501Y SARS-CoV-2 spike variant was less sensitive than either the WT or D614G SARS-CoV-2 spikes. Thus, panels of ACE2 coding variants may be useful for functionally discriminating different SARS-like CoVs, or SARS-CoV-2 variants, which functionally interact with human ACE2 in distinct ways, with each exhibiting a functional "fingerprint" of critical ACE2 residues utilized during entry. This technique may be especially useful for characterizing novel SARS-CoV-2 spike variants, to distinguish those that potentially increase viral fitness through alteration of its ACE2 binding interface, such as N501Y, rather than through other means, such as with D614G. Furthermore, variants like N501Y may affect multiple different steps during infection, and this assay may be useful in characterizing the variants effects on receptor binding in isolation, away from its other effects on immune escape or transmissibility. It is tempting to speculate whether the relative indifference of the SARS-CoV-2 N501Y variant to ACE2 sequence changes may be due to improved adaptation to the human ACE2 sequence, so that infection is less reliant on the properties of singular amino acid side chains at the interface. Moving forward, learning which residues are particularly important for infection may provide more granular details for understanding virus adaptation as the pandemic unfolds.

The cell engineering platform and assays developed in this work can be modified to further interrogate the SARS-CoV-2 spike interaction with ACE2, as well as other diverse virus-host interactions. Importantly, this cell engineering approach is compatible with multiplex assessments of variant effect, permitting the interrogation of thousands of protein variants in pooled assays such as deep mutational scans[18,19,31,48,49]. While a deep mutational scan of ACE2 binding to SARS-CoV-2 spike is published[21], our results highlight differences between ACE2 binding and virus infection, justifying another scan using virus infection as the assay output. Thus, the tools developed in this work could be reused to perform a deep mutational scan comprehensively characterizing how ACE2 sequence changes impact cell entry by SARS-CoV-2 or its variants harboring spike mutations, with cytopathic effect used as the assay readout. Furthermore, using pseudoviruses, the same tools can be harnessed to genetically interrogate virus host interactions relevant for understanding zoonosis, such as assessing how different bat SARS-like coronaviruses engage human ACE2, or even a library of bat ACE2 orthologs. Any viral fusion proteins that can be pseudotyped at efficient titers can be tested in this system. This allows for an in depth characterization of highly pathogenic viruses at reduced biosafety levels, such as Ebolavirus glycoprotein interaction with its host receptor

Niemann-Pick C1[50,51]. Such an application would allow for a broader characterization of species that could carry viruses of public health importance and could help focus ongoing surveying efforts to identify species that have potential for highly pathogenic viral spillover events. These tools may help open up new insights into viral entry, and upon further development, key events during zoonotic viral transmission.

## Materials and methods

### Recombinant DNA construction

LLP-Int-Blast, LLP-iCasp9-Blast, attB-EGFP, and attB-mCherry were described previously [18,19]. All plasmids were created by Gibson assembly[52]. To perform the molecular cloning, a total of 40 ng of plasmid DNA was mixed with 0.333 μM of forward primer, 0.333 μM of reverse primer and 2x Kapa HiFi HotStart ReadyMix (Kapa Biosystems / Roche) added as half the volume. The reaction conditions were 95˚C 5', 98˚C 20", 65˚C 15", 72˚C 5', repeated seven or eight total times, with a final extension for 2' at 72˚C. A total of 1 μL of DPNI enzyme (20 units) was added to each tube, and incubated at 37˚C for 2 h. The reactions were cleaned with a Zymo clean and concentrator kit (Zymo Research). A total of 1 μL of each eluate was mixed together with 1 μL 2x Gibson mix (New England Biolabs) and incubated at 50˚C for 30 min. The recombinant plasmid was transformed into home-made competent *E. coli* 10β cells (New England Biolabs). Plasmids were extracted with a Qiagen or ThermoFisher miniprep kit and sequence-confirmed via Sanger sequencing on an Applied Biosystems 3730 Genetic Analyzer.

The following plasmids were obtained from Addgene: hACE2 was a gift from Hyeryun Choe [2](Addgene plasmid # 1786; http://n2t.net/addgene:1786; RRID:Addgene_1786). psPAX2 was a gift from Didier Trono (Addgene plasmid # 12260; http://n2t.net/addgene:12260; RRID: Addgene_12260). pMD2.G was a gift from Didier Trono (Addgene plasmid # 12259; http://n2t. net/addgene:12259; RRID:Addgene_12259). A pcDNA6-based expression vector for the Wuhan strain of SARS-CoV-2 spike was obtained from the Wilen lab at Yale School of Medicine[44]. pcDNA3-SARS-CoV-2-S-RBD-sfGFP was a gift from Erik Procko[21](Addgene plasmid # 141184; http://n2t.net/addgene:141184; RRID:Addgene_141184). pcDNA3.1-SARS-Spike was a gift from David Veesler [13].

### Cell culture and landing pad clone generation

All cell culture reagents were purchased from ThermoFisher unless otherwise noted. All cell lines were cultured in Dulbecco's modified Eagle's medium supplemented with 10% fetal bovine serum (Corning), 100 U/mL penicillin, and 0.1 mg/mL streptomycin (D10). Cells were passaged by detachment with Trypsin-Ethylenediaminetetraacetic acid 0.25% (Gibco). Landing pad cells were grown in D10 supplemented with 2 μg/mL doxycycline (Fisher), referred to as D10-dox. iCasp9 negative selection was performed with the addition of 10nM AP1903 (ApexBio). Positive selection was performed with 1 μg/mL puromycin (Fisher) for recombined cells or 20 μg/mL blasticidin (Gibco) for long-term passaging of landing pad cells.

Lentivector supernatants for creating Lenti-landing pad lines were produced as previously described[19]. HEK 293T lenti-landing pad lines derived from LLP-Int-BFP-IRES-iCasp9-Blast were generated by incubating HEK 293T cells with various dilutions of lentiviral supernatant ranging from 30 μL to 3 mL. The day after transduction, the media was changed to complete media containing 2 μg/mL doxycycline. At least three days after transduction, the cells were visually confirmed for BFP expression with fluorescence microscopy, and flow cytometry was used to confirm that the cells were transduced at a multiplicity of infection clearly less than one. The cells were then selected with 20 μg / mL blasticidin, and visually confirmed to have the majority of cells die off, further supporting the culture to have been

transduced at a multiplicity of infection less than one. Colonies of clonal cells were picked from the plate and allowed to expand in their own well, and verified for a single integration using a mixture of attB-EGFP and attB-mCherry plasmid as described previously[18,19].

## Recombination of landing pad cells

HEK 293T-based landing pad cells were recombined in either 24-well or 6-well plates, depending on the application. For the 24-well format, 120,000 cells were transfected with 16 ng of pCAG-NLS-Bxb1, a plasmid expressing Bxb1 integrase with a nuclear localization signal to allow for transport into the nucleus, and 238 ng of either individual or a mixture of attB recombination plasmids (or 254 ng of attB recombination plasmid if Bxb1 expression plasmid was not added) incubated with 0.96 μL of Fugene 6 reagent in D10-dox media. For the 6-well format, 600,000 cells were transfected with 80 ng of pCAG-NLS-Bxb1, and 1,120 ng of either individual or a mixture of attB recombination plasmids (or 1,200 ng of attB recombination plasmid if Bxb1 expression plasmid is not added) incubated with 5 μL of Fugene 6 reagent in D10-dox media. Recombined cells were positively selected by growing the cells for a week in D10-dox media supplemented with the indicated amounts of puromycin or blasticidin. Recombined cells were negatively selected with the addition of 10 nM AP1903 / Rimiducid (MedChemExpress). Cells were maintained in AP1903 for 2 days. Recombination rate was assessed by flow cytometry prior to selection.

## Staining for cell-surface ACE2 with SARS-CoV-2 RBD

Soluble SARS-CoV-2 RBD-sfGFP was created by transfecting 1.5 million HEK 293T cells in a 6-well with 1.8 μg of pcDNA3-SARS-CoV-2-S-RBD-sfGFP using PEI-Max MW 40,000 (Poly-Sciences). The media was replaced the next day, and the subsequent media harvested for the next 48 hours. The supernatant was cleared from cells by centrifugation at $300 \times g$ for 3 min, and the soluble fraction retained. ACE2 cell surface staining was performed by removing the ACE2-expressing cells using Versene (Gibco), pelleting the resulting cells at $300 \times g$ for 3 min, and resuspending the cell pellet in supernatant containing SARS-CoV-2 RBD-sfGFP. Following a 30 minute incubation, the cells were pelleted at $300 \times g$ for 3 min, resuspended in PBS + 5% FBS, and immediately analyzed by flow cytometry.

## Lentivector production and infection assays

Lentiviral vectors were produced by transfection of 1.5 million HEK 293T cells in a 6-well plate with 600 ng PsPax2 (Addgene # 12260), 600 ng of the lentiviral transfer vector pLenti_CMV-EGFP-2A-mNeonGreen, and 600 ng of a viral envelope plasmid using PEI-Max MW 40,000 (PolySciences). Media was changed the next day, and the supernatant collected over the next 72 hours. Upon each collection, the media was spun at $300 \times g$ for 3 min, and the soluble fraction retained. For large-scale preps, the supernatant was subsequently passed through a 0.45 μm filter. For small-scale preps of less than 2 mL, the supernatant was used without filtration, avoiding disruption of the cell pellet at the bottom of the tube during pipetting. pMD2.G was used to create VSV-G pseudoviruses, pcDNA3.1(-)SARS-Swt-C9 was used to create SARS-CoV pseudoviruses, and pcDNA6B-SARS2_CoV2-S(Opt)-FLAG or variants thereof were used to create the SARS-CoV-2 pseudoviruses.

## Flow cytometry and fluorescence microscopy

Cells were detached with trypsin, and resuspended in PBS containing 5% serum. Analytical flow cytometry was performed either with a ThermoFisher Attune NxT or a BD LSRII flow

cytometer. For the Attune NxT, mTagBFP2 was excited with a 405 nm laser, and emitted light was collected after passing through a 440/50 nm band pass filter. EGFP was excited with a 488 nm laser, and emitted light was collected after passing through a 530/30nm band pass filter. mCherry was excited with a 561 nm laser, and emitted light was collected after passing through a 620/15 nm band pass filter. iRFP670 and miRFP670 were excited with a 638 nm laser, and emitted light was collected after passing through a 720/30 nm band pass filter. For the BD LSRII, mTagBFP2 was excited with a 405 nm laser, and emitted light was collected after passing through a 440/40 nm band pass filter. EGFP was excited with a 488 nm laser, and emitted light was collected after passing through a B525/50 nm band pass filter. mCherry was excited with a 561 nm laser, and emitted light was collected after passing through a 610/20 nm band pass filter. iRFP670 and miRFP670 were excited with a 640 nm laser, and emitted light was collected after passing through a 710/40 nm band pass filter. Before analysis of fluorescence, live, single cells were gated using FSC-A and SSC-A (for live cells) and FSC-A and FSC-H (for single cells). Fluorescent images were captured on a Nikon Ti-2E fluorescent microscope, equipped with a SOLA SM II 365 light engine (Lumencor), CFI Plan Apochromat DM Lambda 20X objective, DAPI and Texas Red fluorescent filter sets, and imaged with a DS-QI2 monochrome CMOS camera.

## SARS-CoV-2 propagation

The SARS-CoV-2 isolate was obtained through BEI Resources, NIAID, NIH (full designation: SARS-Related Coronavirus 2, Isolate USA-WA1/2020, NR-52281, deposited by the Centers for Disease Control and Prevention). Viral propagation was performed on Vero E6 cells (ATCC CRL-1586) grown in a T175 flask with complete Dulbecco's Minimal Essential Media— (DMEM, GIBCO); 10% heat inactivated Fetal Bovine Serum (FBS, GIBCO); 100 U/mL penicillin, 100 g/mL streptomycin; 2 mM GlutaMAX. On the day of the infection, the SARS-CoV-2 stock was diluted to a MOI of 0.01 in complete DMEM and then added to a confluent T175 flask. Virus adsorption to the cell monolayers was allowed to proceed for 1 hour at 37˚C, 5% $CO_2$ with gentle rocking every 15 min. After adsorption, virus media (DMEM, 2 mM Gluta-MAX, 100 U/mL penicillin, 100 g/mL streptomycin, 100 g/mL non-essential amino acids, and 2% heat-inactivated FBS) was added to the inoculated cells and incubated for 72 hours. Virus recovery was performed by collecting supernatants from cultures when the cytopathic effect involved 50–75% of the cell monolayer and viral supernatant was centrifuged at $300 \times g$ for 10 min at 4˚C to clarify. Viral aliquots were prepared from the clarified supernatant and stored at minus 80˚C until use. All work with infectious SARS-CoV-2 was performed in a Class II Biosafety Cabinet under BSL-3 conditions at Case Western Reserve University School of Medicine, OH, USA.

Viral titer of SARS-CoV-2 was measured by plaque-assay using confluent cell monolayers of Vero E6 cells incubated with serial dilutions of virus for 1 hour at 37˚C, using 24-well plates, with gentle rocking every 15 minutes in virus media. After, each well received an overlay medium containing DMEM with 0.05% agarose and 2% FBS followed by 72 hours incubation at 37˚C. Cells were fixed using a 4% formalin solution in PBS for 30 min, and stained with crystal violet for 5 minutes. Plates were photographed for counting the number of plaques per well, which was used to calculate the viral titer plaque forming units per milliliter (PFU/mL).

## Tissue Culture Infectious Dose 50 (TCID50)

293T-modified cells were grown in T75 flask with selection media: complete DMEM—supplemented with 2 μg/mL doxycycline and 1 μg/mL puromycin. From a confluent flask, cells were

collected, resuspended in selection media, and plated at a final concentration of 10,000 cells per well in a 96-well plate one day before infection. Serial dilutions of SARS-CoV-2 (at a starting titer of 4.0 x $10^5$ PFU/mL on Vero E6 cells) were prepared with selection media. Fifty microliters of virus dilution were added to each well and infection proceeded for 96 hours at 37˚C, 5% $CO_2$. Wells were observed with an inverted microscope to determine which wells showed a cytopathic effect (CPE). The number of CPE positive wells were recorded and used to calculate the median tissue culture infectious dose (TCID50) using the Reed-Muench method[53]. Relative virus quantity was calculated and expressed as TCID50 / mL for each experiment and then normalized to suboptimal Kozak WT ACE2 cells for each individual experiment.

## Western blotting

ACE2-recombined 293T LLP-Int-BFP-IRES-iCasp9-Blast landing pad cells were maintained in D10-dox with 1 μg/mL puromycin. These cells were lysed using commercially procured 1x RIPA buffer (Thermo Scientific, 89901) containing protease inhibitor cocktails (Thermo Scientific, 1862209). Protein lysates were quantified using BCA protein reagent A & B (Thermo Scientific, 23223, 23224) and equal amounts of lysates were denatured in Laemmli buffer followed by separation on 4–12% gradient SDS PAGE gels (Genscript, M00653). Separated proteins on the gel were transferred to a 0.2 μm PVDF membrane (Thermo Scientific, 88520) and immunoblotted using anti- ACE2 (Abcam, Ab15348) and β-actin antibody (Santa Cruz, S47778). Band intensities were assessed using Bio-Rad Image Lab 6.1 software.

## Data analysis and statistics

The large-scale SARS-CoV-2 RBD-sfGFP binding to ACE2 variant data was accessed from the Chan *et al.* Science publication [21]. To aid comparison between the datasets, we rescaled the original Log2 values reported in the original dataset into a fractional binding score. The SARS-CoV RBD binding data was extracted from Fig 3A of the corresponding manuscript [26]. Data analysis was performed using version 1.2.5033 of RStudio, with the exception of flow cytometry data, which was first analyzed for summary statistics using version 10.6.2 of FlowJo. The full list of ACE2 variants tested, as well as human ACE2 variants observed at or near the interface with spike, are described in **S1 Table**. The pseudovirus infection values used for the analyses in the manuscript are recorded in **S2 Table**. The primers used to generate the tested ACE2 variants are found in **S3 Table**. The data, as well as an R Markdown file containing the code for the full analysis have been uploaded to the Matreyek Lab GitHub repository at http://github.com/MatreyekLab/ACE2_variants.

Statistical tests for differences in pseudovirus infection rates were performed using a two-sample T-test, which was corrected for multiple hypothesis testing using the Benjamini-Hochberg procedure[54], and only samples with p-values smaller than 0.01 were labeled on the plots as significant. The code for the statistical analysis is recorded and can be recreated using the R Markdown file in the aforementioned GitHub repository.

## Supporting information

**S1 Fig. Full set of ACE2 expression constructs that were tested.** A) Schematics describing each ACE2 expression construct. IRES, Internal Ribosome Entry Site; H2A, Histone 2A; Pac, Puromycin N-acetyltransferase. B) Wide-field fluorescent imaging of ACE2 overexpressing cells, with nuclei stained with Hoechst 33342 (left) and intracellular mCherry distribution (right). C) Representative single-cell distributions of red mean fluorescence intensities, as captured by flow cytometry, in cells left uninduced (orange) or induced to express ACE2 from the

Tet-inducible promoter using 2 µM doxycycline (black). D) Replicate staining of cell-surface ACE2 using SARS-CoV-2 RBD-sfGFP. Vertical bars denote the means calculated from the geometric means from each individual replicate. N = 2 and 8 for uninduced and induced cells, respectively. Error bars denote 95% confidence intervals. E) Percentages of mCherry+ cells that also stain for cell-surface ACE2, using SARS-CoV-2 RBD-sfGFP. N = 5, error bars denote 95% confidence intervals.
(TIF)

**S2 Fig. Pseudovirus infection assays.** A) Schematic showing a self-inactivated, third-generation lentiviral vector encoding GFP behind a CMV promoter. B) Percentage of green+ cells detected with various dilutions of SARS-CoV spike -pseudotyped green lentiviral vector inoculum. Data from three independent replicates are shown. The gray bar denotes a slope of 1.5. C) Pairwise comparisons of titers derived from equal inocula of SARS-CoV and SARS-CoV-2 spike-pseudotyped lentiviruses, from three independent replicates. N = 3. D) Fold infection in cells expressing each of the indicated ACE2 expression constructs, normalized to infection of parental HEK 293T cells. N = 3, error bars denote 95% confidence intervals. E) Representative smooth histograms of EGFP mean fluorescence intensity observed with flow cytometry. Blue corresponds to EGFP translated behind a consensus "GCCACC" Kozak sequence, orange corresponds to EGFP translated behind the suboptimal "CATTGT" Kozak sequence, and red corresponds to background green fluorescence of the parental cell line in the absence of EGFP.
(TIF)

**S3 Fig. Quantitation of ACE2 abundance with consensus and suboptimal Kozaks as well as endogenous expression from reference samples.** A) To improve quantitation of the relative fold-difference in ACE2 protein expression between the consensus and suboptimal Kozak cells, we electrophoresed small quantities of cell lysates from consensus Kozak ACE2 cells, adding unmodified 293T lysate to normalize for overall protein content, and compared them to undiluted cell lysates from suboptimal Kozak ACE2 cells. B) Western blotting of full-length ACE2 in modified 293T and Vero E6 cells. C) Quantitation of endogenous ACE2 protein abundance in 293T and Vero E6 cells as compared to the suboptimal Kozak ACE2 transgenic cells. The black horizontal line denotes geometric means of replicate blots, and error bars denote 95% confidence intervals. D) Histogram of ACE2 transcripts per million observed in various tissues (purple) or primary cell types (orange) catalogued in GTEx. HEK 293T cells had a 0.1 ACE2 TPM. TPM values corresponding to the relative increase in protein abundance observed in suboptimal Kozak ACE2 transgenic cells (0.4) and Vero E6 cells (1.6) are shown as the dotted blue and red lines, respectively.
(TIF)

**S4 Fig. Infection of cells encoding ACE2 germline variants outside of the interaction surface.** A) Infection of a panel of ACE2 variants of residues outside of the spike interaction interface, which were observed in publicly available human genomics datasets. B) Western blot quantitation of recombinant cells expressing WT ACE2 or its variants behind the suboptimal Kozak. Short horizontal lines are the geometric mean of three independent replicates, and the error bars denote 95% confidence intervals from three replicates. C) Scatterplots comparing WT and Actin-normalized ACE2 Western blotting band intensity to SARS-CoV and SARS-CoV-2 pseudovirus infection results. D) Relative pseudovirus infection rates (y-axis) and published binding to SARS-CoV RBD as assessed by immunoprecipitation (x-axis), for WT ACE2 and its variants. E) Relative pseudovirus infection rates (y-axis) and the value of the published SARS-CoV-2 RBD-sfGFP staining (x-axis) for WT ACE2 and its variants, comparing either the values from the inverse of the low SARS-CoV-2 RBD-sfGFP binding population (nCoV-

S-Low; left) or the high binding population (nCoV-S-High; right) sorts.
(TIF)

**S5 Fig. Raw Western blotting images used for quantitation.** Raw, unscaled western blotting exposures for three replicate experiments assessing the steady-state abundances of the key ACE2 variants tested in the study, blotting for ACE2 protein shown in the top row and blotting for beta-actin as a loading control shown on the bottom row.
(TIF)

**S6 Fig. Alignment of ACE2 N-terminal sequences in multiple bat species.** The human ACE2 sequence was used as the query in a Blast search, and ACE2 sequences from the *Rhinolophus* bat species were collected, with representative members shown in an alignment. The position corresponding to K31 in humans is shown in the partially transparent yellow box. The human ACE2 sequence is shown for comparison.
(TIF)

**S1 Table. Table of variants tested, as well as observed human ACE2 variants near the SARS-CoV-2 spike interface.** Descriptions of column titles: pmid—Pubmed ID of publication describing control variants; rationale—rationale for why the variant was included in the study; position—amino acid positions in the ACE2 protein; snv—"yes" indicates that the variant can be created through single nucleotide variation from the human ACE2 sequence; _allele_freq— allele frequency of the variant in the database, _homozygotes—number of homozygous samples of the variant found in the database; _hemizygotes—number of hemizygous males with the variant found in the database; sars_cov2_infection—geometric mean of the normalized SARS-CoV-2 pseudovirus infection values observed in our experiments.
(TSV)

**S2 Table. Summary of pseudovirus infection data generated in this work.** Descriptions of column titles: recombined_construct—Shorthand identifier for the plasmid used for recombination; cell_label—label used in figures; date—the date flow cytometry was performed, either in YYMMDD or MM/DD/YY notation; expt—experiment category the sample belongs to; pseudovirus_env—which viral envelope was used; pseudovirus_inoc—the number of uL of pseudovirus prep used in the sample; live_singlets—number of cells that passed the FSC-A/ SSC-A and FSC-A/FSC-H gates and were considered live singlets; pct_grn—percent of cells that were green within the live singlest population; pct_red—percent of cells that were red within the live singlest population; pct_grn_gvn_red—percent of cells that were green within the red cell population; moi—multiplicity of infection; scaled_infection—moi of the sample, normalized to the ACE2(low) negative control; log_scaled_infection—the normalized infection that was log10 transformed for subsequent analysis.
(TSV)

**S3 Table. List of primers used to create variant constructs used in this study.**
(DOCX)

## Acknowledgments

We wish to thank Simone Edelheit and Milena Zelembaba of the Genomics Core Facility of the CWRU School of Medicine's Genetics and Genome Sciences Department, Mike Sramkoski, Kyla Johnson, and D'Arbra Blankenship of the CWRU Cytometry & Imaging Microscopy Core Facility of the Case Comprehensive Cancer Center, and Sophia Onwuzulike of the CWRU BSL3 facility, for maintaining key research equipment and resources integral to this work.

## Author Contributions

**Conceptualization:** Anna M. Bruchez, Kenneth A. Matreyek.

**Data curation:** Kenneth A. Matreyek.

**Formal analysis:** Kenneth A. Matreyek.

**Funding acquisition:** Kenneth A. Matreyek.

**Investigation:** Nidhi Shukla, Sarah M. Roelle, Vinicius G. Suzart, Anna M. Bruchez, Kenneth A. Matreyek.

**Methodology:** Anna M. Bruchez, Kenneth A. Matreyek.

**Project administration:** Kenneth A. Matreyek.

**Resources:** Kenneth A. Matreyek.

**Supervision:** Anna M. Bruchez, Kenneth A. Matreyek.

**Visualization:** Kenneth A. Matreyek.

**Writing – original draft:** Kenneth A. Matreyek.

**Writing – review & editing:** Nidhi Shukla, Anna M. Bruchez, Kenneth A. Matreyek.

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
