## [Decision Letter · Decision Letter 0]

7 May 2021

Dear Dr. Matreyek,

Thank you very much for submitting your manuscript "Human ACE2 variants differentially promote SARS-CoV and SARS-CoV-2 spike mediated infection" for consideration at PLOS Pathogens.

Your manuscript has been reviewed by two relevant experts, both of whom had a largely favorable evaluation of the work. Most of their suggestions seem constructive and relatively easily addressable, so please do your best to address the feasible ones in a revised version, and if any cannot be addressed in revision please explain why in a reviewer response that you submit with the revised manuscript.

Sincerely,

Jesse D Bloom, Ph.D.

Guest Editor

PLOS Pathogens

Ron Fouchier

Section Editor

PLOS Pathogens

Kasturi Haldar

Editor-in-Chief

PLOS Pathogens

orcid.org/0000-0001-5065-158X

Michael Malim

Editor-in-Chief

PLOS Pathogens

orcid.org/0000-0002-7699-2064

Your manuscript has been reviewed by two relevant experts, both of whom had a largely favorable evaluation of the work. Most of their suggestions seem constructive and relatively easily addressable, so please do your best to address the feasible ones in a revised version, and if any cannot be addressed in revision please explain why in a reviewer response that you submit with the revised manuscript.

Reviewer Comments (if any, and for reference):

Reviewer's Responses to Questions

**Part I - Summary**

Reviewer #1: Shukla et al. combine cell line engineering and studies of cellular infection to understand how SARS-CoV-2 spike-mediated cellular infection is quantitatively related to ACE2 expression and affinity for the spike RBD. Furthermore, they survey how rare ACE2 variants found in humans impact susceptibility to cellular infection. They show quite strikingly that ACE2 overexpression masks detrimental effects of ACE2 variants on cellular infection. They construct a cell line with reduced ACE2 expression that sensitizes measurements of ACE2 variation, identifying rare variants that hinder SARS-CoV and SARS-CoV-2 entry, and some variants that differentially effect infection by SARS-CoV, SARS-CoV-2, and SARS-CoV-2+N501Y. In several figures, they show quite nice “threshold-like” relationships between ACE2:RBD binding affinity or ACE2 expression with susceptibility to cellular infection. These results are important in large part because ACE2-overexpresing cell lines are used so widely for various questions about SARS-CoV-2 biology. The results also report technological advances for cell line engineering that enable future work as nicely described in the Discussion.

Reviewer #2: The manuscript by Shukla et al. described a procedure in which ACE2 gene was inserted into a specific location in the genomic DNA of 293T cells so that the cells constitutively express ACE2 as a receptor for SARS-CoV and SARS-CoV-2. It further analyzed how the ACE2 variants from human populations support the cell entry of SARS-CoV and SARS-CoV-2. Overall, the concepts of engineering a site-specific integration of ACE2 gene into the cellular genome and analyzing the coronavirus receptor activities of naturally existent ACE2 variants are interesting. I have the following comments.

Major:

(1) After reading the title, I was expecting some interesting analysis on how certain human populations are more resistant or more susceptible to SARS-CoV-2 infections, only to find out that this manuscript, like many other studies in the literature, was mainly about how some ACE2 mutations affect the proteins's coronavirus receptor activities. Specifically, the finding correlating the allele frequencies of the ACE2 variants and their coronavirus receptor activities, which is supposed to be most interesting point of the study, were presented in a small panel as Figure 4E. So could the authors summarize these data in a more comprehensive table or figure so that people can evaluate the risk of being infected by SARS-CoV-2 for certain populations?

(2) Could the authors clarify how the allele frequencies of ACE2 variants were determined and how accurate these numbers are? Also, What is the rationale or criteria for choosing the 18 out of 276 known ACE2 variants for testing in the current study?

(3) The authors made a lot of negative comments on other assays that study coronavirus/receptor interactions. However, the current manuscript mainly relies on pseudovirus entry assay and flow cytometry, both of which are less accurate than some other biochemical assays (such as surface plasmon resonance). The authors need to tone down these statements.

(4) The authors repetitively claimed that their approaches are the most physiologically relevant because of the multivalent binding interactions between multiple spike molecules and multiple ACE2 molecules, but over-expressing ACE2 in cells is quite different from what happens in vivo (ACE2 is known to be lowly expressed in the upper respiratory tracts that SARS-CoV-2 targets). The authors need to tone down these statements.

Minor:

(1) In the introduction, the authors forgot to include furin as one of the proteases that activate SARS-CoV-2.

(2) It seems that the authors measured the ACE2 expression in cell lysates. But to quantify the cell surface expression of ACE2, cell membrane-associated ACE2 needs to be measured.

(3) Some of the cited literatures are inaccurate.

(4) Isn't data a plural?

(5) Line 148: Purified recombinant RBD should be used, not supernatants containing the RBD.

(6) Cells are "transducted" by pseudovirus particles, not "challenged". There are also several other inaccurate virology terms used in the manuscript.

(7) Do these ACE2 mutations affect the peptidase activities of ACE2?

(8) The discussion is lengthy. It needs to be shortened and focus on relevant topics.

(9) The results section also needs to be cleaned up to be more concise.

**Part II – Major Issues: Key Experiments Required for Acceptance**

Reviewer #1: n/a

Reviewer #2: (No Response)

**Part III – Minor Issues: Editorial and Data Presentation Modifications**

Reviewer #1: The main suggestion that I have that is a genuine critique, is that I would be left more satisfied if I had a bit more contextualization of what “relevant” ACE2 expression levels might be. The authors several times simply refer to a general lack of knowledge on this question – which is perhaps true and is not something I have looked into myself. But my impression (again, I fully admit this is not something I have looked for or found in actual papers), is that tissue-level ACE2 expression is thought to be ‘relatively low’. I understand that’s not something you can hang a hat on, but I guess I just want to emphasize, if there are any straws of physiological relevance that can be grasped, it would help a lot in understanding whether I should truly ‘care’ about the ACE2-low results more than the highly overexpressed cell line where impacts on cell entry are masked.

One place I can think to point toward for discussing “physiological relevance” is studies on antibody neutralization. For example, in a recent preprint (https://doi.org/10.1101/2021.04.03.438258), the antibody S309 is shown to have incomplete and reduced neutralization in ACE2-high cell lines compared to cell lines expressing lower levels of ACE2. S309 demonstrates efficacy in animal models (Fc-independent) as well as protective efficacy in its clinical trial as VIR-7831 (though this in theory could be neutralization-independent functions, though neutralization is still probably at play). This should be confirmed in the literature, but I believe antibodies to the spike NTD similarly do not neutralize (?) in ACE2-high cell lines despite neutralizing in ACE2-low cell lines and protecting in animal models. These observations would therefore suggest that the ACE2-low cell lines were more “representative” of in vivo and could potentially be discussed.

Alternatively (and I only suggest this because I see Vero E6 cells are already at the authors’ disposal per the methods), would it be helpful to detect ACE2 levels in Vero E6 cells by Western in comparison to the ACE2-engineered cell lines? Since this would be a ‘native cell’ expression level to which to compare engineered cells (though I do not know the history of Vero E6 cells and if its ACE2 expression level would be expected to be at least somewhat representative of the original tissue).

Some other minor comments:

1. I had some difficulty parsing the one Figure Legend I tried to parse. Fig. 1D-F: I think figure legend text or annotation on plot is missing for explaining the difference between induced and uninduced cells as black versus yellow coloring? 1G legend incomplete: geometric mean of what? (Once again, can deduce but the axis titles maybe aren’t super obvious to all readers.) 1H legend, “Fold infection of ACE2 expression” typo? I didn’t read the remaining legends as carefully (as it is clear what is happening in each figure alongside the text – which is great), but probably all figure legends should be checked for similar levels of self-sufficiency and clarity)

2. Starting line 227: these ‘variants’ that are being discussed – by comparison to the paragraph starting on line 236, my impression is that these are not human variants, but rather specifically chosen mutations – perhaps from structural reasons? In my reading of the paper, I was led to think at first this meant these were human variants. Perhaps if they are not, they should just be referred to as ‘mutants’.

3. Is there any estimate of how the spike density on PV relates to density on SARS2 virions? Perhaps these results suggests that expression from the spike side of the coin is also important to think about in future study?

4. This is totally an aside/out of scope, but the last paragraph made me think about long-term technology development – another really powerful development would be if it were possible to not only create libraries with different receptor orthologs or variants, but also somehow combine this with libraries on the viral side, for library-on-library assays in the context of full cellular infection. I’m not sure if this could be managed with some sort of second landing pad site that the viral genome can integrate into, and/or some single-cell / droplet-based microbiology to link viral and receptor identifier barcodes. Anyway, that would obviously be a very long-term endeavor, but would be a very cool direction!

Reviewer #2: (No Response)

PLOS authors have the option to publish the peer review history of their article (what does this mean?). If published, this will include your full peer review and any attached files.

Reviewer #1: No

Reviewer #2: No

Figure Files:

Data Requirements:

Reproducibility:

References:

---

## [Editor Report · Decision Letter 1]

15 Jun 2021

Dear Dr. Matreyek,

We are pleased to inform you that your manuscript 'Variants of human ACE2 differentially promote SARS-CoV and SARS-CoV-2 spike mediated infection' has been provisionally accepted for publication in PLOS Pathogens.

Please *also note that the editor has requested two more small revisions when you submit the final version.*

Best regards,

Jesse D Bloom, Ph.D.

Guest Editor

PLOS Pathogens

Ron Fouchier

Section Editor

PLOS Pathogens

Kasturi Haldar

Editor-in-Chief

PLOS Pathogens

orcid.org/0000-0001-5065-158X

Michael Malim

Editor-in-Chief

PLOS Pathogens

orcid.org/0000-0002-7699-2064

EDITOR COMMENTS: 

The reviewer comments have been addressed, and the new data seem solid. I am recommending acceptance of the manuscript, and do not see any need to send out it for more review as the original reviews were already favorable and the critiques are fully addressed. However, I have two further minor critiques listed below that I ask the authors to address when they upload a final version:

1. I still think the abstract and title need to better reflect that most of the ACE2 "variants" are engineered mutants and not natural alleles in the human population. This is clear in the Results, but I think the abstract / title may still mislead some readers here. Maybe "mutation" is a better word than "variant" here, or the authors can make some other edit to capture the spirit of this concern.

2. Line 150: “accurate” should be removed as it’s not 100% clear pseudoviruses capture all relevant features of actual viral entry.
---

## [Editor Report · Acceptance letter]

2 Jul 2021

Dear Dr. Matreyek,

We are delighted to inform you that your manuscript, "Mutants of human ACE2 differentially promote SARS-CoV and SARS-CoV-2 spike mediated infection," has been formally accepted for publication in PLOS Pathogens.

Best regards,

Kasturi Haldar

Editor-in-Chief

PLOS Pathogens

orcid.org/0000-0001-5065-158X

Michael Malim

Editor-in-Chief

PLOS Pathogens

orcid.org/0000-0002-7699-2064